# Engineering the Electronic Structure towards Visible Lights Photocatalysis of CaTiO_3_ Perovskites by Cation (La/Ce)-Anion (N/S) Co-Doping: A First-Principles Study

**DOI:** 10.3390/molecules28207134

**Published:** 2023-10-17

**Authors:** Qiankai Zhang, Yang Wang, Yonggang Jia, Wenchao Yan, Qinghao Li, Jun Zhou, Kai Wu

**Affiliations:** 1School of Electronics and Information, Xi’an Polytechnic University, Xi’an 710048, China; 2Xi’an Key Laboratory of Interconnected Sensing and Intelligent Diagnosis for Electrical Equipment, Xi’an Polytechnic University, Xi’an 710048, China; 3State Key Laboratory of Electrical Insulation and Power Equipment, Xi’an Jiaotong University, Xi’an 710049, China

**Keywords:** CaTiO_3_, photocatalysis, first-principles calculations, co-doping, electronic structure

## Abstract

Cation-anion co-doping has proven to be an effective method of improving the photocatalytic performances of CaTiO_3_ perovskites. In this regard, (La/Ce-N/S) co-doped CaTiO_3_ models were investigated for the first time using first-principles calculations based on a supercell of 2 × 2 × 2 with La/Ce concentrations of 0.125, 0.25, and 0.375. The energy band structure, density of states, charge differential density, electron-hole effective masses, optical properties, and the water redox potential were calculated for various models. According to our results, (La-S)-doped CaTiO_3_ with a doping ratio of 0.25 (LCOS1-0.25) has superior photocatalytic hydrolysis properties due to the synergistic performances of its narrow band gap, fast carrier mobility, and superb ability to absorb visible light. Apart from the reduction of the band gap, the introduction of intermediate energy levels by La and Ce within the band gap also facilitates the transition of excited electrons from valence to the conduction band. Our calculations and findings provide theoretical insights and solid predictions for discovering CaTiO_3_ perovskites with excellent photocatalysis performances.

## 1. Introduction

Recently, extensive research has been conducted on the use of photocatalysts with superior water splitting activity to produce hydrogen [1,2] in order to meet the new energy demands [3] and to resolve the environmental issues caused by traditional fossil fuels [4]. In this regard, the CaTiO_3_ (CTO) perovskite has gained considerable attention for its broad photocatalytic properties, including its ability to degrade organic pollutants and reduce carbon dioxide emissions in an environmentally friendly manner [5,6,7]. However, pure CaTiO_3_ exhibits only moderate photocatalytic activity for hydrogen evolution due to its large band gap (~3.5 eV), which is only capable of using part of ultraviolet light, accounting for only 5% of solar energy [8]. On the one hand, the choice of preparation methods [9], including solid-state, co-precipitation, mechanochemical, sol-gel, hydrothermal and solvothermal process, significantly impacts the properties of perovskite materials Among these methods, hydrothermal routes for CaTiO_3_ have gained prominence due to their ability to produce well-crystallized nanoparticles with customizable shape and size at relatively low processing temperatures [10]. On the other hand, a variety of techniques have been applied to engineer the band structure of CaTiO_3_ to enhance its photocatalytic capacity by extending its absorbance spectrum to visible light, such as element doping [11], heterojunction/nanocomposite construction [12], structural/phase control [13], and perovskite derivatives [14]. Notably, element doping has attracted considerable attention as a research focus due to its streamlined implementation and meticulous regulation, which aim to reduce the band gap and enhance the photocatalytic efficiency of CaTiO_3_ [15,16,17].

The mono-doping technique is one of the simplest and earliest methods of improving the photocatalytic performance of CaTiO_3_ by adding cations or anions to its crystal lattice [18,19]. Anion doping in CaTiO_3_ results in variations in its valance band structure, thereby directly affecting its absorbance spectrum. For instance, Wang et al. [20] demonstrated that nitrogen (N) doping at the oxygen (O) site could extend the absorption edge into the visible region and reduce the band gap, resulting in enhanced photocatalytic activity. A similar strategy to manage the electronic structure of oxide perovskites by sulfur (S) doping has proven to be an effective approach to achieve catalytic improvement in energy storage and conversion [21]. In such cases, element doping, specifically with higher atomic energy containing 2p and 3p orbitals, at the O site could elevate the valence band maximum (VBM) of the host catalyst, thereby reducing the bandgap [22]. First-principles calculations have also provided theoretical support for the proposition that substitutional N doping at O sites introduces impurity states within the band gap [23]. In addition, Attou et al. [24] conducted density functional theory (DFT) calculations to investigate the doping effect by replacing oxygen sites in CaTiO_3_ with boron(B) revealing that the appearance of the p energy level of dopants leads to a narrowed band gap and enhanced response to visible light. 

In contrast to the upshift of the VBM caused by anion doping in CaTiO_3_, the introduction of cations to the CaTiO_3_ crystal lattice will result in substantial changes in the conduction band minimum (CBM) or the appearance of impurity energy levels in the band gap, which could also affect its photocatalytic efficiencies. For this reason, metallic cations have been widely used as dopants in CaTiO_3_ to improve its photocatalytic properties. It has been found that the introduction of vanadium (V) into the titanium (Ti) site in CaTiO_3_ effectively enhances photocatalytic activity by creating impurity energy levels of V just below the conduction band edge [25]. Moreover, the introduction of copper (Cu) dopants at the Ti sites within the CaTiO_3_ crystal leads to notable enhancements in photocatalytic activities, particularly under visible light irradiation, which can be attributed to the transition from the donor level created by the presence of Cu impurities at the conduction band of CaTiO_3_ [26]. However, the doping of mental elements at the Ti site may result in the conversion of Ti^4+^ to Ti^3+^ and the formation of oxygen vacancies [27]. In a similar manner, calcium (Ca) site-doped cations can also be used to fine-tune the electronic energy levels of perovskite materials, thereby affecting their band gaps, conductivities, and optical properties [28]. As demonstrated by Anzai et al. [29], a moderate amount of lanthanide (La) doping at the Ca site in the crystal lattice led to commendable photocatalytic efficiency under visible light. The impurity energy levels generated within the band gap as a result of metal element doping serve as either donor or acceptor energy levels, facilitating the utilization of visible light. When an excited electron transition is step-assisted by the donor or acceptor energy levels in the band gap, it narrows the band gap, allowing visible light to be absorbed [30]. It is important to note, however, that while the mono-doping of cations or anions in CaTiO_3_ perovskites has demonstrated beneficial effects on its photocatalytic performance, it may also result in lattice defects in the crystal. These defects may facilitate the recombination of photogenerated electrons and holes, thereby destabilizing the sample [31]. Thus, doping strategies must be carefully considered and optimized in order to strike a balance between photocatalytic activity and unwanted side effects.

In order to have a more delicate modification of the electronic structure of CaTiO_3_ perovskites and minimize its side impacts on the crystal lattice, the co-doping strategy [32] gradually emerged as a prevailing method for enhancing its visible-light responsiveness, including different ion pairs such as cation–anion [33,34], cation–cation [35,36,37], or anion-anion [38,39,40]. Cation-cation co-doping and anion-anion co-doping techniques provide more precise control of CBMs and VBMs, respectively, than mono-doping [41]. In contrast, cation-anion co-doping offers a synergistic treatment of the band structure by modulating CBMs with cations and VBMs with anions, in addition to the occurrence of impurity energy levels in the band gap. For example, Sulaeman et al. [42] conducted experimental investigations that validated a significant enhancement in both the light absorption capacity and the photocatalytic performance of La/N co-doped perovskite. This enhancement was achieved through the substitution of Ca site cations by La^3+^ and O^2−^ by N^3−^, respectively. In a separate study, the band gap was significantly reduced through co-doping with N–La, carbon–celerium (C-Ce), and N–Ce–N, as predicted by DFT calculations [43]. These cation-anion co-doped models mentioned above also displayed superior crystal lattice stability in comparison to their mono-doped counterparts, thus mitigating the formation of photogenerated carrier recombination centers. Similarly, Devi et al. [44] synthesized Ce/N, S co-doped catalysts, which exhibited higher photocatalytic activities under both UV and visible light conditions. This can also be attributed to the occurrences of Ce^3+^ dopant energy levels below the CBM and the N^3−^ acceptor energy levels above the VBM.

It is evident from the statement above that rare earth ions are excellent candidates for regulating the band structure due to their incompletely occupied 4f and empty 5d states. Besides, the synergistic effect of donor energy levels and nonmetals provided by La^3+^/Ce^3+^ and N^3−^/S^2−^ may be an effective enhancement of the photocatalytic performance of CaTiO_3_. Therefore, co-doping of rare earth metal cations and anions has found significant applications in catalytic materials, particularly in perovskites. However, to the best of our knowledge, this particular pair of cation-anion dopants, La/Ce-N/S, has never been studied previously for CaTiO_3_ both experimentally and theoretically, but are believed to be promising and effective dopant candidates for broadening the absorbance spectrum and enhancing the photocatalytic properties of perovskites. In light of this, we present a comprehensive investigation of the bulk, electrical, and optical properties of (La/Ce-N/S) co-doped CaTiO_3_ using DFT calculations. We constructed a 2 × 2 × 2 supercell, consisting of 40 atoms, to investigate the effects of La/Ce and N/S co-doping and the microscopic influence of doping concentration on the structural phase and electronic structure of CaTiO_3_. Through a comprehensive evaluation of formation energy and binding energy for various doping configurations, we identified two optimal models, namely LCOS (La-S co-doped CaTiO_3_) and CCOS (Ce-S co-doped CaTiO_3_). These models were subsequently selected for in-depth investigation regarding the concentration (0.125, 0.25, 0.375) of rare earth metal elements. Concurrently, we conducted calculations pertaining to the energy band structure, density of states, charge differential density, and electron-hole effective masses within the model. Furthermore, the light absorption spectrum and water-splitting potential were analyzed. This study aims to demonstrate the potential of band structure engineering through the co-doping of rare earth and non-metal cations as a promising strategy for enhancing the activity of photocatalysts. 

## 2. Results and Discussion

### 2.1. Computational Methods

The density functional theory (DFT) method is an accurate method for calculations of the energetics and electronic structures of solids that used in this study. We conducted all calculations using CASTEP, a powerful quantum calculation module in Materials Studio based on the DFT theory plane-wave pseudopotential method for exploring the electronic and structural properties of crystals such as minerals, metals, and semiconductors [45]. In the field of electronic calculations for CTO perovskites, it is one of the most popular choices [46,47]. Electronic interaction energies were computed employing the spin-polarized generalized gradient approximation (GGA) functional and the Perdew–Burke–Ernzerhof (PBE) functional [48]. In the Brillouin zone, the cutoff energy for fundamental functions was set to 571.4 eV, and the k-point sampling was 3 × 3 × 3. The interactions involving ion nuclei and valence electrons were elucidated using ultrasoft pseudopotentials. To approximate the experimental band gap of 3.50 eV in CaTiO_3_ [49], the GGA+U approach was implemented to account for correlation effects. In computations, the conditions were set to Ueff = 4.3 eV, 8.1 eV, and 7.0 eV to describe the Ti 3d, La 4f, and Ce 4f states, respectively [50]. Regarding the convergence criteria, the energy was constrained to 1.0 × 10^−5^ eV/atom, the maximum force on atoms did not exceed 3.0 × 10^−2^ eV/Å, the maximum stress on atoms was limited to 5.0 × 10^−2^ GPa, and atom displacements were restricted to 1.0 × 10^−3^ Å.

### 2.2. Crystal Structure and Formation Energy

In this study, we selected the highly crystalline CaTiO_3_ perovskite (space group Pm-3m), which comprises a twelve-coordinated Ca atom, a four-coordinated Ti atom, and a two-coordinated O atom. To introduce rare earth elements (specifically lanthanum and cesium), we substituted the calcium atom in the bulk structure at a doping ratio of 0.125. In addition, oxygen atoms in the lattice were replaced with low electronegativity atoms as part of a common doping approach. As a result of the hybridization of the electron orbitals of the dopant elements with those of the O 2p states, new molecular orbitals were formed that had a lower energy than it. Consequently, the VBM derived from these new molecular orbitals was higher than that of the O 2p states. To assess the impact of co-doping, we selected two elements with low electronegativity (nitrogen and sulfur). We constructed a 2 × 2 × 2 supercell, consisting of 40 atoms, to investigate the effects of rare earth element co-doping and the microscopic influence of doping concentration on the structural phase and electronic structure of CaTiO_3_ (see Figure 1). As the majority of perovskite materials employ this model structure in their calculations to maintain precision [8,47,51], our choice of this model enhances the reasonableness and reliability of our subsequent calculations. Moreover, the doping effect was initiated at different positions during co-doping by replacing oxygen atoms proximally and distally (1-proximally and 2-disrally), depending on the position of doped rare earth elements.

The optimized lattice parameters of pure CaTiO_3_ (a = b = c = 3.899 Å) were in agreement with both experimental findings (a = b = c = 3.897 Å) [52] and theoretical data (a = b = c = 3.931 Å) [53]. The calculated Ti–O bond length was determined to be 1.950 Å, while the closest interatomic separation between Ca and O was computed at 2.758 Å. These values closely aligned with experimental measurements of 1.952 Å for Ti–O and 2.760 Å for Ca–O, respectively [54]. In this regard, the calculation method and parameters used for modeling calcium titanate were appropriate. Given the comparable ionic radii of N^3−^ and O^2−^, the incorporation of N dopant into the lattice induced subtle deviations in the lattice parameters; nevertheless, the observed alterations were deemed negligible in magnitude. Conversely, due to the larger radius of S^2−^ compared to O^2−^, the lattice volume significantly increased, regardless of whether the doping occurred at the proximal or distal end. By comparing the formation energy and binding energy of co-doping, the validity of the calculation can be determined.

The formation energy ($E_{f}$) was employed to assess the feasibility of dopant integration and the stability of the doped system within the lattice, as described by Equation (1) [46]:(1)$$E_{f}=(E_{doped}-\sum n_{doped}\mu_{doped})-(E_{pure}-\sum n_{superseded}\mu_{superseded})$$
where $E_{doped}$ and $E_{pure}$ are the total energy of doped and pure CaTiO_3_ systems, respectively. $n_{doped}$ and $n_{superseded}$ represent the number of doped and superseded elements. $\mu_{doped}$ and $\mu_{superseded}$ refer to the chemical potential of doped and superseded elements.

For metal elements, $\mu_{Ca}$, $\mu_{Ti}$, $\mu_{La}$, and $\mu_{Ce}$ are defined as the total energy of metal divided by the number of atoms in bulk. For non-metallic elements, $\mu_{o}$, $\mu_{S}$, and $\mu_{N}$ are obtained by putting the most stable element in a 10 Å × 10 Å × 10 Å cube box, respectively. For fluctuating chemical potential during dopant production, computations were undertaken in O-rich and O-poor environments, according to Equation (2) [47]: (2)$$\frac{\left(\mu_{CaTiO_{3}}-\mu_{Ca-bulk}-\mu_{Ti-bulk}\right)}{3}\leq \mu_{o}\leq \mu_{o-gas}$$
where $\mu_{CaTiO3}$ represents the total energy of pure CaTiO_3_. The superior and inferior boundaries of $\mu_{o}$ are classified as O-rich and O-poor environments, respectively. The formation energy for O-rich ($E_{f}^{Orich}$) and O-poor ($E_{f}^{Opoor}$) cases can be calculated by plugging the appropriate $\mu_{o}$ into Equation (1). The formation energy of co-doped CaTiO_3_ in O-rich and O-poor conditions was summarized in Table 1.

As defined by the definition of formation energy, structures with smaller formation energies would be more stable and the process of doping would be easier. As shown in Table 1, the formation energy was more negative in an O-poor environment than in an O-rich environment, indicating thermodynamic stability. The difference in formation energy led to the easiest doping site. Also, regardless of changes in rare earth elements (La and Ce), N was more predisposed to a remote site (N2), whereas S was more appropriate for doping close to rare earth elements (S1). In addition to the analysis of doping formation, calculating the binding energy ($E_{b}$) of the doping model was essential to determine whether the system was more tightly bound (Equation (3)) [55]:(3)$$E_{b}=E_{X-doped}+E_{Y-doped}-E_{pure}-E_{(X+Y)-doped}$$
where $E_{X(Y)-doped}$ represents single doping energy and $E_{X+Y-doped}$ refers to the co-doped energy. The binding energy of co-doped CaTiO_3_ is summarized in Table 2. It is evident that higher positive binding energies correspond to more well-defined doping models. Therefore, both binding energy and formation energy had similar trends when both rare earth elements (La or Ce) and low-electronegativity elements (N or S) were simultaneously incorporated (Table 2). Specifically, the S atom is more likely to preferentially occupy the proximal position, while the N atom is expected to predominantly reside in the distal position.

Therefore, four typical co-doping systems (LCON2, LCOS1, CCON2, and CCOS1) were selected for subsequent investigations. A comparison between N and S doping revealed that the introduction of S atoms favored the formation of the rare earth co-doping model even though it may have a larger lattice constant. Therefore, we examined the LCOS1 and CCOS1 models, focusing on the effects of the rare earth element concentration. The formation energy of Ca_x_La_1−x_TiO_2.95_S_0.05_ (x = 0.25, 0.375) and Ca_x_Ce_1−x_TiO_2.95_S_0.05_ (x = 0.25, 0.375) are displayed in Table 3. For comparison, the energies of La/Ce single doping were calculated under the same conditions. Increasing the doping ratio results in favorable formation energies for the La/Ce single-doped models, whereas adding S in an oxygen-deficient environment generates more negative formation energies for the co-doped systems. This supports the notion that the co-doped systems are more thermodynamically favorable.

### 2.3. Electronic Structure

To reveal the changes in the electronic structure of CaTiO_3_ after co-doping, DFT calculations were performed on pure CaTiO_3_, four typical co-doping systems, and four co-doping systems with different rare earth concentrations. The band structure of CaTiO_3_ is depicted in Figure 2.

The widely recognized band gap inaccuracies inherent in DFT computations yielded a calculated band gap of 2.2 eV for pure CaTiO_3_ employing the GGA+U approach (Figure 2). This value remains 1.3 eV lower than the experimentally determined band gap width of CaTiO_3_. Nonetheless, the band structure characteristics and the shift in the band gap remain consistent and reliable when compared to prior calculations [56,57,58].

In the case of N atom doping, a modest shift in the band gap was observed, as depicted in Figure 3a,b. In general, doping elements with lower electronegativity than oxygen tend to maintain the conduction band minimum (CBM) while elevating the top of the valence band. However, the introduction of rare earth elements (La and Ce) had only a marginal impact on the band gap position, which decreased to 2.15 eV and 2.14 eV, respectively, as the co-doped system approached equilibrium. By contrast to the co-doping of N and rare earth elements, the co-doping of S with La or Ce introduced a distinct intermediate band near the Fermi level (Figure 3c,d). It is worth noting that the presence of S alone may not be the sole reason for the formation of the intermediate band (IB). The origin of the IB can be attributed to the introduction of La^3+^ and Ce^3+^ ions. Since the superseded ion is Ca^2+^, the excess electrons lead to the formation of donor energy levels. Conversely, the co-doping of N^3−^ compensates for the electron states, resulting in the absence of intermediate energy levels in the band gap. On the other hand, the co-doping of S, which belongs to the same main group as oxygen, exhibits the presence of an intermediate energy band. The hybridization between the S orbital and O orbital seemed very insufficient due to the lower electronegativity of S. As a result, the S atomic orbital did not significantly contribute to the formation of the valence band and the upward shift of its maximum. It effectively conveys the conclusion that the intermediate band is not solely attributed to the presence of non-metallic elements or rare earth elements alone, but rather to the change in energy level induced by their co-doping. 

Figure 3e,h compares the changes in the band gap of the co-doped model at different doping concentrations. In the presence of La and Ce ions, there is an increase in impurity energy levels, which alters the model of the unit cell. To comprehensively compare the influence of various proportions of rare earth elements, the band structure was also calculated for single doping of 0.25 and 0.375 La/Ce, as shown in Appendix A. From Appendix A, it is evident that a high doping concentration results in a significant number of intermediate band gaps. This N-type doping facilitates the excitation of electrons from the intermediate band gap to the conduction band, enhancing the separation of photogenerated carriers. This finding further confirms that the formation of intermediate levels is not solely attributed to the single doping of rare earth or non-metallic elements.

Additionally, at a doping concentration of 0.25, the model exhibits the narrowest band gap, indicating that proper doping of rare earth elements can regulate the band gap width of CaTiO_3_ and enhance its visible light absorption and carrier migration. On the other hand, combining rare earth metal elements with non-metals (S) results in a widening of the intermediate band gap with increasing concentration. However, unlike single doping, co-doping causes only a modest increase in the band gap, which may lead to electron-hole recombination issues. Under Ce doping, a high concentration of 0.375 causes the impurity energy level to cross the Fermi level, slightly lowering the bottom of the conduction band, and possibly resulting in half-filled impurity energy band electrons. In contrast, spin-up and spin-down impurity energy levels under a high concentration of La doping appear close to the Fermi level. The impurity levels shift below the Fermi energy level, occupying filled states, which is beneficial for reducing complex centers [59]. Based on the combination of single doping and co-doping, it can be concluded that a high concentration of Ce and S co-doping causes the intermediate level to cross the Fermi level and leads to a half-filled state, exacerbating electron-hole recombination and subsequently affecting the efficiency of photocatalytic water splitting. To further analyze the distribution of orbital energy levels, the density of states (DOS) of multiple atoms with different orbitals was investigated.

The DOS data of pure CaTiO_3_ are depicted in Figure 4. The Ti 3d and O 2p states composed the conduction band and valence band of pure CaTiO_3_, respectively. These data agreed well with the previous simulations [53,58]. A minor tail was created at the Fermi level due to the smearing of the calculation settings, without an actual impact.

Unlike pure CTO, the introduction of La/Ce and N/S elements influences the band distribution to varying degrees, as shown in Figure 5. This effect is particularly evident in O 2p states and Ti 3d states. In Figure 5a,b, the substitutional doping of La and Ce does not exhibit a discernible effect near the Fermi level. The N 2p and O 2p states show a slight hybridization at the top of the valence band, with the O 2p states remaining as the dominant contributors. In Figure 5c,d, a minor peak is observed at the Fermi level, primarily composed of Ti 3d states and O 2p states, as confirmed by the inset. This indicates that S does not directly contribute to the generation of impurity energy levels during doping, which aligns with the findings in the energy bands.

Furthermore, an increase in the doping concentration of rare earth elements leads to changes in the composition and intensity of impurity energy levels. Similarly to the previous conclusion, higher doping concentrations of La/Ce have a minimal direct impact on the energy band, as the substitutional Ca atoms do not participate in the formation of the conduction and valence bands. However, higher doping concentrations result in impurity levels appearing closer to the Fermi level (Figure 5e–h). This phenomenon can be attributed to the lattice distortion caused by rare earth and S elements, which in turn influences the Ti 3d states and O 2p states [60]. A comparable trend is observed in the single doping of rare earth elements, as depicted in Appendix A. The intermediate band gap near the Fermi level is composed of Ti 3d states and O 2p states, with the predominance of Ti 3d states varying with the doping concentration. Moreover, the increase in impurity levels filled with electrons facilitates the movement of electrons from the valence band to the conduction band, thereby enabling transitions to the bottom of the conduction band at lower energy through the intermediate band. Consequently, this phenomenon holds the potential to enhance the efficiency of photocatalytic water splitting.

When the doping concentration is 0.25, the S 3p state slightly emerges in the component of the impurity level. At a concentration of 0.375, the dominant contribution to the energy level is still accounted for by the O 2p and Ti 3d states. The La/S-doped model depicted in Figure 5g exhibits both spin-up and spin-down energy levels. The magnetic moment analysis indicates that the magnetic moments of all co-doped models near the Fermi level are predominantly contributed by Ti, which aligns with the PDOS results. In contrast, Ce doping results in a partial magnetic moment, with the total magnetic moment of the model reaching a maximum value of 7 μ_b_ at a concentration of 0.375. The impurity energy level crosses the Fermi energy level, resulting in a no longer full band energy level [47].

Consequently, this significantly reduces the light absorption capacity and promotes the formation of recombination centers. Combining the band structure and PDOS analysis, it can be concluded that the introduction of impurity energy levels into the model through co-doping facilitates the migration of photogenerated carriers and improves catalytic activity. With the increasing La/Ce doping concentration, the impurity level also increases; however, the Ce-doped impurity level no longer constitutes a full-electrical band [59]. 

The electron density differences were calculated to explore the influence of doping on charge transfer and interaction. The difference in electron density between pure CTO, co-doped perovskite, and heavily doped perovskite is represented in Figure 6.

Here, the (0, −1, 1) crystal plane is utilized to demonstrate the charge transfer of doping elements in greater detail. In the CTO crystal lattice, there is a significant charge density between Ti and O, indicating a strong interaction between Ti and O. On the other hand, the interaction between Ca and O appears to be less pronounced, suggesting a weaker contact between Ca and O. The bonding between Ca and O tends to be ionic, while the Ti-O bonds exhibit more covalent characteristics [8]. In the case of N doping (Figure 6b,c), noticeable covalent bonding features are displayed. This can be attributed to the hybridization between the N 2p and O 2p states, resulting in the formation of the top of the valence band as observed in the above calculations.

Similar to N-doping, S-doping also leads to the formation of covalent bonds to a certain extent. However, due to the lower electronegativity of S compared to O and N, the bonding effect is diminished, resulting in minimal hybridization between the S 3p state and the O 2p state at the top of the valence band. La/Ce doping yields two distinct outcomes. In the case of Ce doping, there is a higher electron concentration around Ce, causing electrons around O to be closer to the Ce atom regardless of the changes in doping concentration [51]. On the other hand, La doping did not participate in the transfer of electrons. As illustrated in Figure 6d–i, both rare earth elements and co-doping affect the aggregation of electrons around Ti, indirectly influencing the energy level changes in the model. The Ti 3d state predominantly contributes to the generation of impurity energy levels at the center of the band gap, consistent with Figure 5c–h. 

We also conducted a comprehensive Bader charge analysis [61] on all models in order to determine an atom’s charge state quantitatively by comparing its calculated charge with its own valence charge (Table 4, Appendix A). In the pristine model, the results align well with the electron density difference shown in Figure 6, where Ca and Ti exhibit electron loss, establishing ionic and covalent bonds with O. The addition of S to the O site in the LCOS1-0.25 model results in Ti distortion due to its lower electronegativity and larger atomic size. Ti atoms adjacent to S atoms exhibit varying degrees of electron loss as compared to pure CTO. Despite the fact that this distortion affects the charge transfer process between neighboring atoms, it remains relatively stable. Similarly, in the pristine CTO model, electrons appear to be uniformly distributed at each type of atom. The average Bader net atomic charge for Ca, O, and Ti is 1.33, −1.04, and 1.80, respectively. The Bader net charge distribution is significantly imbalanced following the introduction of 0.25 concentrations of La and S doping, with that of Ca, O, and Ti varying from 0.48 to 1.47, −1.52 to −0.25, and 1.47 to 1.90, respectively. In addition, the average Bader net atomic charge for Ca and Ti decreases to 0.90 and 1.70, respectively, while the average Bader net atomic charge for O increases to −0.92. This non-uniform charge distribution leads to structural distortions within the model, which can potentially result in the formation of impurity levels [62]. Our findings are in line with our earlier calculations, which indicated that La and S co-doping resulted in impurity energy levels below the Fermi level. These impurity energy levels act as energy steps within the band gap, facilitating the transition of photogenerated carriers.

It should be noted that Bader net atomic charges for S-doped sites are considerably lower than those for O-doped sites, which confirms that hybridization is limited for their contributions to VBM and impurity levels [63]. This is consistent with our DOS analysis in Figure 5 and the electron density difference in Figure 6. Consequently, the co-doped model alters the electron density around Ti as the concentration of rare earth elements increases, and higher concentrations of La have little effect on the lattice, thus promoting the stability and photocatalytic efficiency of the CaTiO_3_ perovskite. 

### 2.4. Effective Masses of Charge Carriers

Light excitation below the Fermi level results in the generation of excited electrons at the bottom of the conduction band and holes at the top of the valence band, which are crucial for desired chemical processes. To determine whether doping affects the separation of electrons and holes, many works have focused on the mobility of photogenerated carriers. However, accurately calculating their mobility is challenging as it requires considering various scattering mechanisms, including intrinsic factors like electron-phonon scattering and extrinsic factors like impurity scattering [64]. Hence, estimating the effective mass of electrons and holes can offer valuable insights into the migration of photogenerated carriers. 

When electrons and holes possess a low effective mass, they exhibit greater activity in their excited state and can more swiftly migrate to active sites [65]. To explore the transfer properties of photogenerated carriers, the effective masses of electrons (m_e*_) and holes (m_h*_) in pure CaTiO_3_, La/Ce-doped CaTiO_3_, and (La/Ce + N/S)-codopedCaTiO_3_ were estimated by performing a parabolic fit to the CBM and VBM using the following equation [41,66].
(4)$$m^{*}=\hbar^{2}(d^{2}E/dk^{2}{)}^{-1}$$
where *m** represents the effective masses of carriers, *ħ* is the Planck constant and d^2^*E*/d*k*^2^ is the coefficient fitting the second-order term of the E(*k*) curve for the base of the conduction band and the top of the valence band. The effective masses of electrons and holes, determined along the most dispersive direction in the Brillouin zone, and their corresponding k-paths are provided in Table 4. It should be noted that the CCOS1-0.375 model could not be calculated due to its half-filled intermediate band (IB) crossing the Fermi level, resulting in the semiconductor transitioning into a conductor. As presented in Table 5, the computed effective masses of electrons and holes in pure CTO are 0.069 m_e_ and 0.084 m_e_, respectively. Notably, the electron in the conduction band exhibits a smaller effective mass compared to the hole in the valence band. Upon single doping with varying proportions of rare earth elements, it becomes evident that the masses of electrons and holes at a concentration of 0.25 are considerably smaller than those at a concentration of 0.375. This indicates enhanced mobility of electrons and holes, facilitating the rapid movement of carriers to the catalytic surface during the photoexcitation process.

Moreover, similar observations can be made regarding the intermediate band. Since the intermediate band gap is fully occupied below the Fermi level, its likelihood of acting as a recombination center is substantially reduced. Nevertheless, the effective mass of carriers within the intermediate band also plays a crucial role in determining photocatalytic efficiency. As the intermediate band gap serves as a “springboard”, it is essential for electrons to possess high mobility to enable swift transitions into the conduction band. Additionally, when both rare earth and non-metallic elements are doped, the effective mass of electron-hole pairs increases, particularly in the case of S doping. This can be attributed to the dominance of the O/S state, which is susceptible to disorder at the chalcogen site at the top of the valence band [64]. Notably, when the concentration of rare earth elements is raised to 0.25, the effective mass of electrons and intermediate band holes decreases. For instance, the effective mass of electrons and intermediate band carriers in the LCOS1-0.25 model are 0.050 m_e_ and 0.087 m_e_, respectively. These findings align with the conclusions drawn from the single doping of rare earth elements. Co-doping rare earth and non-metal elements at a concentration of 0.25 promotes the mobility of photogenerated carriers, thus enhancing photocatalytic efficiency.

### 2.5. Optical Properties and Water Redox Potential

The photocatalytic properties of materials are greatly influenced by their optical absorption range. Therefore, the optical absorption spectra of different models, including pure CaTiO_3_ and co-doped variants with various mass fractions, were examined. To assess the light absorption properties, the dielectric function was converted into a bit-light absorption coefficient using Equation (5) [67].
(5)$$\alpha_{abs}=\sqrt{2}\omega \sqrt{\sqrt{\varepsilon_{1}^{2}\left(\omega \right)+\varepsilon_{2}^{2}\left(\omega \right)}-\varepsilon_{1}\left(\omega \right)}$$
where *ω* represents the phonon angular frequency, and ε_1_(*ω*) and ε_2_(*ω*) are the real and imaginary parts of the complex of dielectric constants, respectively. Furthermore, the imaginary part characterizes light absorption capability, while the real part signifies electron excitation energy. Despite employing the GGA+U method for electronic structure correction, it does not impact systematic bandgap width investigation. Consequently, underestimating the bandgap leads to significant errors in optical property predictions during light absorption. 

To address this, we adopted the “scissors” operator, a widely recognized strategy for rectifying such inaccuracies [34]. In this calculation, the scissors operator was set at 1.3 eV, determined by the Band Gap Correction method, which is a commonly used method for calculating the value of the scissors operator by contrasting experimental and theoretical bandgap values, thereby increasing prediction accuracy for the optical property. To be more precise, the value of the scissors operator is obtained by subtracting the experimental value (~3.5 eV) [68,69,70] from its theoretical value (2.2 eV). The calculated optical absorption curves are provided in Figure 7 and Appendix A. The modified absorption range of pure CTO correlated well with previous experiments [37,71]. Moreover, the fact that pure CTO can only absorb ultraviolet light within a narrow range severely limits the photocatalytic activity.

Moreover, we can analyze the differences in light absorption of different doping models in conjunction with the electron density difference in Figure 6 and the effective mass of electrons and holes in Table 4. As a result of the hybridization between the O 2p and Ti 3d states [72], the bonding between Ca and [TiO_6_] can be described as ionic, whereas the bonding between Ti and O can be described as covalent. This is also in accord with our calculated PDOS results. In the presence of increased doping concentrations, the charge transfer at Ti atoms undergoes significant changes, resulting in distortions of the crystal structure as well as conversions between [TiO_6_] and [TiO_5_] [73]. The band gap and the edges of the light absorption spectrum may be affected by these changes [74]. These results can be attributed to electron excitations and transitions [75], where electrons move from the valence band to the impurity energy levels and then subsequently to the conduction band of CaTiO_3_, which also aligns well with our earlier calculations. The impurity energy levels occur for those co-doping models with 0.25 concentration, along with a more substantial reduction in the band gap and the lowest electron-hole effective mass. A leftward shift of the light absorption peak can be observed as the doping concentration increases. Figure 7 illustrates that the introduction of a single rare earth element through doping substantially enhanced the visible light absorption range of CTO. At a concentration approaching 0.25, the strongest visible light absorption peak (464 nm), corresponding to the smallest band gap value, was observed. This shift is primarily due to the increased band gap and the introduction of more intermediate band gap states by La/Ce [76]. Consequently, electrons tend to be more localized at the impurity levels, impeding the material’s light absorption properties.

In the low-wavelength phase (150–300 nm), as shown in Appendix A, we can clearly observe a decreasing trend in light absorption within our model framework, whereas pure CTO exhibits an increase at around 200 nm [77]. Conversely, in the high-wavelength stage (700–900 nm), the absorption spectrum of pristine CTO is entirely absent [78]. In spite of the gradual decrease in absorption intensity, the rare earth doping model at 0.25 concentration still maintains its light absorption capability. The above statement provides valuable theoretical guidance regarding the study of light absorption at various doping concentrations of rare elements in CTO perovskite.

Further, the light absorption edge of CCOS1-0.375 was extended to 800 nm, due to its crystal structure distortions caused by high doping concentrations. The strong optical absorption in this region is primarily due to the transition of electrons from the valence band to the conduction band of the models. As predicted by our previous calculations, the high-concentration model showed an increased presence of impurity levels and a comparatively smaller band gap. However, impurity energy levels intersecting the Fermi level would act as recombination centers for photogenerated carriers, thereby reducing its photocatalytic efficiency. In spite of the substantial expansion of the light absorption range at high concentrations, the large electron-hole mass and the fully occupied impurity levels below the Fermi level continue to restrain the photocatalytic performance of the model.

As compared to pure and low-concentration systems, the photoabsorption spectrum of highly concentrated doped CaTiO_3_ exhibits a noticeable redshift, with new absorption peaks at 406 nm for LCOS1-0.375. It’s worth noting that the robust optical absorption in the visible region primarily arises from electron transitions from their VBM to CBM [75]. As previously calculated, the high-concentration model showed an increase in impurity levels and a comparatively reduced band gap. The results were attributed to the emergence of full-filled impurity levels below the Fermi level, which allowed more electrons to be excited to the conduction band. Despite the minor enhancement in light absorption intensity, similar to single doping, we contend that the LCOS1-0.25 model is more effective at absorbing and transporting visible light-excited electrons to the sample surface due to its band gap and electron-hole mass analysis as previously discussed. Although the visible light absorption intensity of co-doping was lower than that of single doping, it is much higher than that of pure CTO, thereby improving the photocatalytic efficiency. 

The position of the semiconductor band edge is a crucial signal for determining the water-splitting capability and performance of semiconductor photocatalysts. This factor is necessary for CBM of the semiconductor photocatalysts to become more negative than the reduction potential of H^+^/H^2^ (0 eV vs. normal hydrogen electrode (NHE)), as well as for VBM to become more positive than the oxidation potential of O_2_/H_2_O (1.23 eV) [26]. Therefore, by correcting the scissors operator, the CBM and VBM were calculated empirically according to Equation (6) [60]: (6)$$E_{CB}=X-\frac{1}{2}E_{g}-E_{e}$$
where $E_{CB}$ represents the bottom level of the conduction band, *X* is the absolute electronegativity of the perovskite oxide catalyst, $E_{g}$ refers to the corrected band gap, and $E_{e}$ is the energy of free electrons on the hydrogen scale (4.5 eV). Further calculations for the VBM can be performed using the corrected band gap and CBM from Equation (5). The obtained data are visualized in Figure 8.

As depicted in Figure 8, photoexcitation caused electrons to transition from the valence band to the conduction band, and the presence of impurity levels reduced the transition energy, which is advantageous for enhancing photocatalytic performance. Furthermore, the VBM and CBM positions of calcium titanate (CTO) spanned the water redox potential level, meeting the thermodynamic requirements for hydrogen synthesis in sunlight-driven water splitting. However, a wide bandgap of 3.50 eV hindered the movement of photogenerated carriers in pure CTO. In the case of single doping, the system exhibited energy bands and visible light absorption intensity that satisfied the requirements of the hydrolyzed band edge, particularly at a concentration of 0.25. However, the presence of impurity levels at this concentration may act as carrier recombination centers. The calculation of the effective mass of electrons and holes indicated that LCTO-0.25 exhibited high mobility.

In contrast to N doping, co-doping of S and rare earth elements introduced impurity energy levels that fulfilled the water redox potential with a smaller band gap, corresponding to the previously estimated energy band and DOS (Appendix A). This phenomenon can be attributed to the presence of impurity bands at the Fermi level originating from the Ti 3d and O 2p states, where Ti attracted more electrons based on charge density analysis. Furthermore, with an increase in the concentration of rare earth doping, the bandgap initially decreased and then increased. LCOS1-0.375 exhibited better performance in the visible light spectrum due to its VBM being closer to the water splitting potential and having a higher density of impurity levels compared to the narrowest bandgap observed with a La doping ratio of 0.25 (estimated to be 2.26 eV). For Ce doping, the bandgap was also significantly reduced by high-concentration doping, reaching 2.37 eV at a doping concentration of 0.25. However, further increasing the doping concentration seemed to impact the semiconductor properties (Figure 6). More electrons surrounded Ce, potentially affecting the photocatalytic performance.

Regarding the co-doped system, the energy required for the transition from the valence band to the impurity level was lower, facilitating electron transition. Considering the effective mass of the photogenerated carrier, LCOS1-0.25 exhibited higher carrier mobility and satisfied the minimum hydrolyzed band edge energy, thereby improving the photocatalytic efficiency of CTO. Therefore, the co-doping of La/Ce and S effectively narrowed the bandgap and adjusted the positions of the VBM and CBM by introducing impurity levels. Notably, with increasing La doping proportion, La/S enhanced the photocatalytic performance of CaTiO_3_ photocatalysts, aligning with the conclusions drawn from single doping. Combining the energy bands, carrier mobility, and visible light absorption, LCTO-0.25 and LCOS1-0.25 exhibited the strongest photocatalytic performances. According to the calculations in Table 6, a systematic comparison of different doping strategies on photocatalytic properties of different perovskite-based photocatalysts was listed. Despite having a reduced band gap and an extended visible light absorption range, Pd-doped models still suffer from recombination of photogenerated carriers owing to a relatively smaller effective mass for electrons and holes compared to the La-S co-doped model. Furthermore, the simultaneous introduction of La and S into CaTiO_3_ created more intermixed impurity energy levels within the band gap, which has also increased its photocatalytic effectiveness. The systematic theoretical analysis of the different doping strategies summarized in Table 6 offers valuable insight into future research on CTO perovskites, particularly in terms of assessing their overall photocatalytic performance.

## 3. Conclusions

In summary, CaTiO_3_ perovskite with the doping of rare earth elements (La/Ce) and N/S elements was investigated using the GGA-PBE functional. Co-doping rare earth elements and S had a significant impact on the energy band structure of CTO, leading to changes in the band gap and electronic transition energy through the introduction of impurity energy levels. La doping at increased concentrations elevated the impurity energy level, resulting in enhanced performance in the visible light spectrum. However, higher doping concentrations, such as 0.375, led to the emergence of impurity energy levels crossing the Fermi level, which acted as recombination centers, diminishing photocatalytic efficiency. PDOS and electron density differences analysis confirmed the presence of impurity states induced by co-doping, which altered the hybridization orbitals between Ti 3d and O 2p states. The effective mass calculations provided insights into the mobility of photogenerated carriers, revealing that co-doping rare earth and non-metal elements at a concentration of 0.25 exhibited higher carrier mobility and satisfied the minimum hydrolyzed band edge energy requirements for efficient photocatalysis. Additionally, optical absorption spectra analysis showed a redshift and broadening of photoabsorption peaks in co-doped CTO materials, indicating a smaller band gap and improved light absorption performance. The introduction of impurity energy levels by doping elements contributed to reduced transition energy and enhanced photocatalytic efficiency. Our findings suggest that doping CTO with rare earth and non-metal elements is a promising strategy for tailoring electronic structure, enhancing optical absorption range, and improving photocatalytic performance.

## Figures and Tables

**Figure 1 molecules-28-07134-f001:**
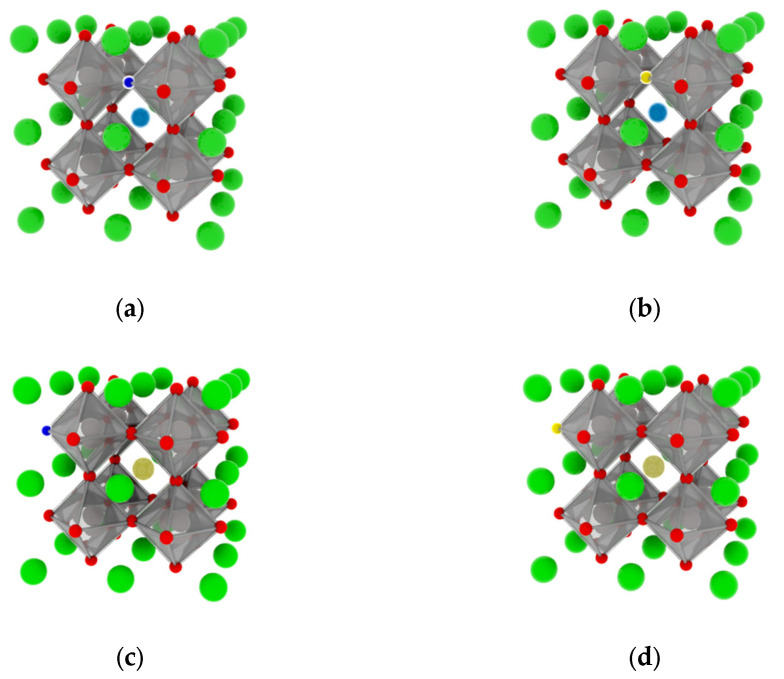
Crystal structures: (**a**) La_0.125_Ca_0.875_TiO_2.95_N_0.05_ (LCON1), (**b**) La_0.125_Ca_0.875_TiO_2.95_S_0.05_ (LCOS1), (**c**) Ce_0.125_Ca_0.875_TiO_2.95_N_0.05_ (CCON2), and (**d**) Ce_0.125_Ca_0.875_TiO_2.95_S_0.05_ (CCOS2). (Calcium atoms-green, Titanium atoms-gray, oxygen atoms-red, Lanthanum atoms-dark blue, Cesium atoms-dark yellow, Nitrogen atoms-light blue, and Sulfur atoms-light yellow).

**Figure 2 molecules-28-07134-f002:**
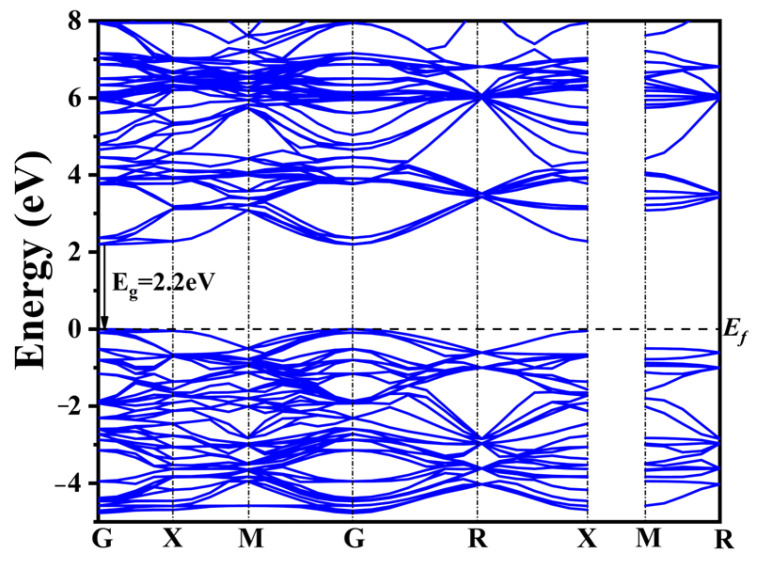
Band structure of pure CaTiO_3_.

**Figure 3 molecules-28-07134-f003:**
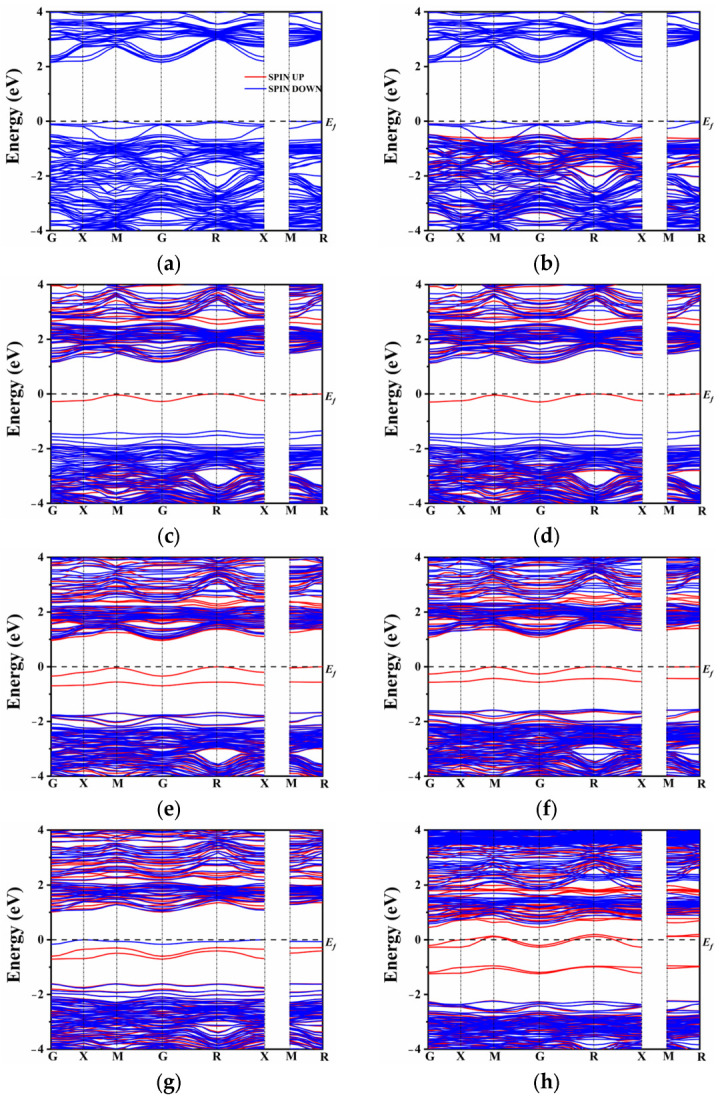
Band structure: (**a**) LCON2, (**b**) CCON2, (**c**) LCOS1, (**d**) CCOS1, (**e**) LCOS1-0.25, (**f**) CCOS1-0.25, (**g**) LCOS1-0.375, and (**h**) CCOS1-0.375.

**Figure 4 molecules-28-07134-f004:**
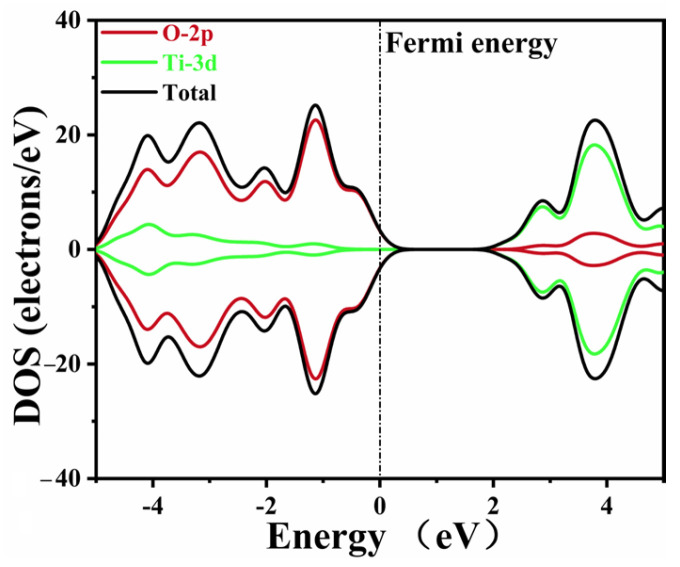
The density of states of pure CaTiO_3_.

**Figure 5 molecules-28-07134-f005:**
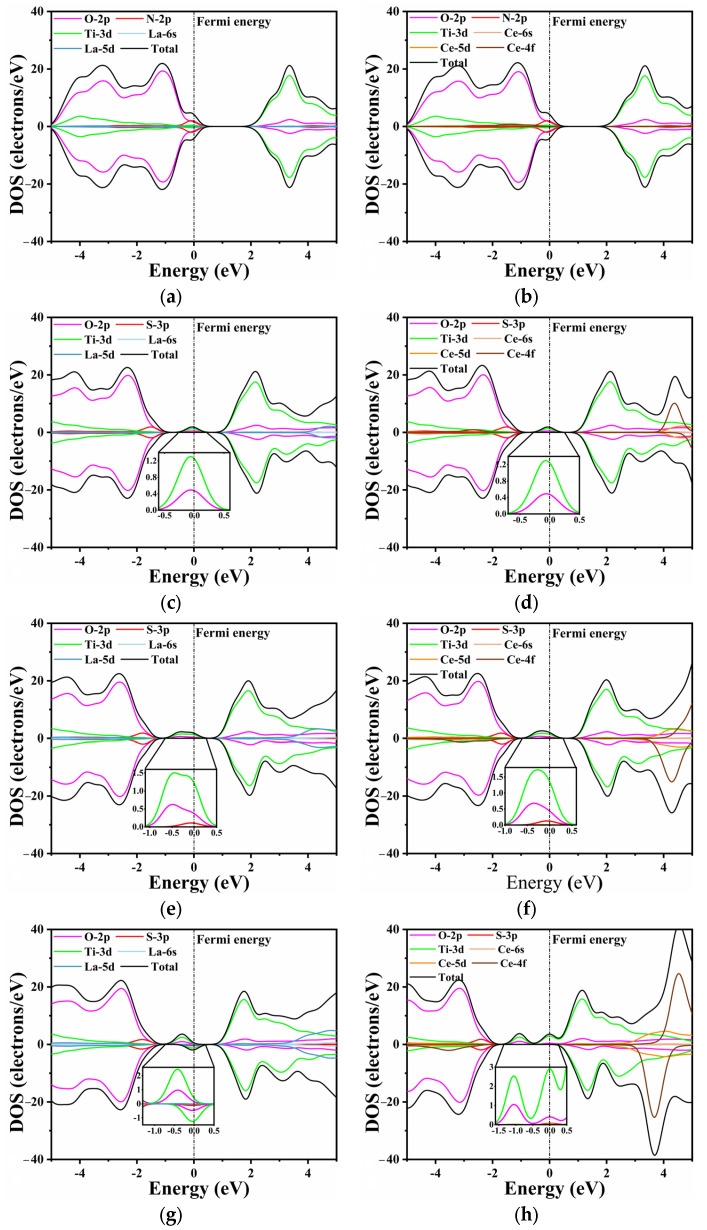
The TDOS and PDOS: (**a**) LCON2, (**b**) CCON2, (**c**) LCOS1, (**d**) CCOS1, (**e**) LCOS1-0.25, (**f**) CCOS1-0.25, (**g**) LCOS1-0.375, and (**h**) CCOS1-0.375. The inset depicts impurity band orbitals corresponding to PDOS, respectively.

**Figure 6 molecules-28-07134-f006:**
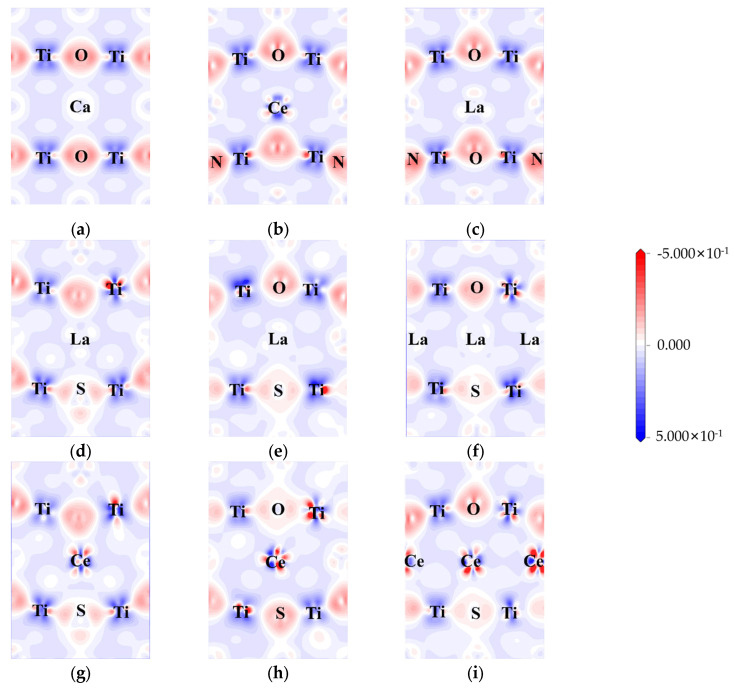
Electron density difference: (**a**) CTO, (**b**) CCON2, (**c**) LCON2, (**d**) LCOS1, (**e**) LCOS1-0.25, (**f**) LCOS1-0.375, (**g**) CCOS1, (**h**) CCOS1-0.25, and (**i**) CCOS1-0.375. (The electron density difference maps are along (0, −1, 1) crystal plane).

**Figure 7 molecules-28-07134-f007:**
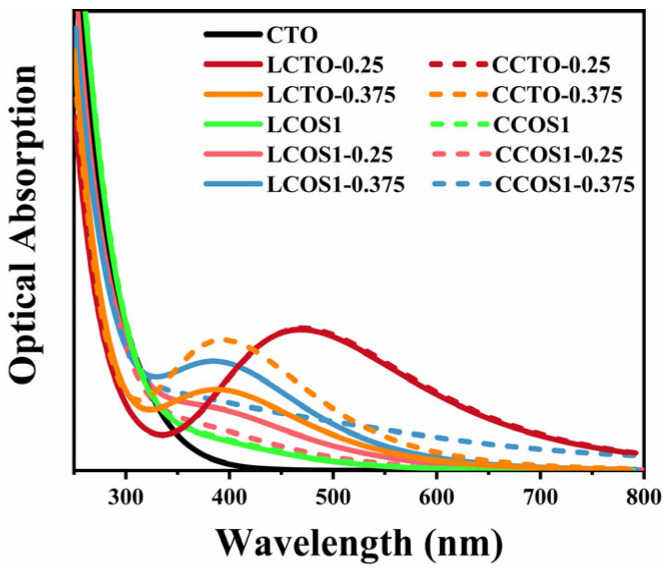
Optical absorptions of pure and doped CTO.

**Figure 8 molecules-28-07134-f008:**
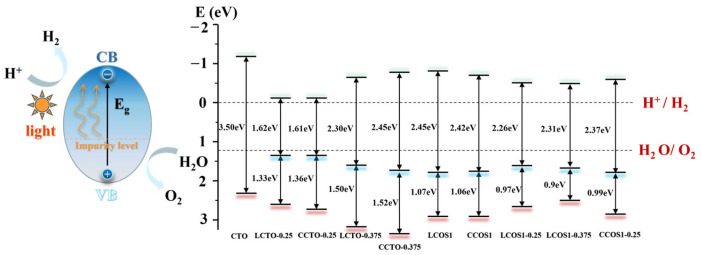
The CTO−based photocatalytic water splitting for hydrogen generation and water splitting potential analysis.

**Table 1 molecules-28-07134-t001:** The formation energy of the co-doped model.

Co-Doped Model	$E_{f}^{Orich}$(eV)	$E_{f}^{Opoor}$(eV)	Co-Doped Model	$E_{f}^{Orich}$(eV)	$E_{f}^{Opoor}$(eV)
LCON1	1.318	−4.427	CCON1	1.828	−3.917
LCON2	0.947	−4.797	CCON2	1.800	−3.944
LCOS1	0.841	−4.904	CCOS1	1.718	−4.026
LCOS2	1.360	−4.385	CCOS2	1.845	−3.900

**Table 2 molecules-28-07134-t002:** The binding energy of the co-doped model.

Co-Doped Model	$E_{b}$ (eV)	Co-Doped Model	$E_{b}$ (eV)
LCON1	0.793	CCON1	1.378
LCON2	1.251	CCON2	1.494
LCOS1	3.375	CCOS1	3.593
LCOS2	2.856	CCOS2	3.467

**Table 3 molecules-28-07134-t003:** The formation energy and binding energy of the co-doped and single doped model.

Co-Doped Model	$E_{f}^{Orich}$(eV)	$E_{f}^{Opoor}$(eV)	Doped Model	$E_{f}$ (eV)
LCOS1-0.25	0.237	−5.508	LCTO-0.25	−0.920
LCOS1-0.375	−0.428	−6.173	LCTO-0.375	−4.470
CCOS1-0.25	1.979	−3.766	CCTO-0.25	1.005
CCOS1-0.375	2.298	−3.447	CCTO-0.375	−1.952

**Table 4 molecules-28-07134-t004:** The calculated Bader charge for CaTiO_3_ and LCOS1-0.25.

Model	CaTiO_3_		LCOS1-0.25	
Number	Atom Type	Bader Net Atomic Charge	Atom Type	Bader Net Atomic Charge
1	Ca	1.33	Ca	1.47
2	Ca	1.33	Ca	0.48
3	Ca	1.33	Ca	0.55
4	**Ca**	**1.33**	**La**	**1.38**
5	Ca	1.33	Ca	1.43
6	Ca	1.33	Ca	0.91
7	Ca	1.33	Ca	0.76
8	**Ca**	**1.33**	**La**	**1.09**
9	O	−1.05	O	−0.60
10	O	−1.05	O	−0.67
11	O	−1.04	O	−0.83
12	O	−1.05	O	−0.71
13	O	−1.05	O	−0.87
14	O	−1.05	O	−0.70
15	O	−1.05	O	−0.84
16	O	−1.04	O	−0.96
17	O	−1.05	O	−0.97
18	O	−1.04	O	−0.66
19	O	−1.04	O	−0.46
20	O	−1.05	O	−0.26
21	O	−1.05	O	−1.38
22	O	−1.04	O	−1.24
23	O	−1.04	O	−1.32
24	O	−1.04	O	−1.45
25	O	−1.04	O	−1.52
26	O	−1.04	O	−1.22
27	O	−1.04	O	−1.51
28	O	−1.04	O	−0.44
29	O	−1.04	O	−0.60
30	**O**	**−1.04**	**S**	**−0.62**
31	O	−1.04	O	−0.82
32	O	−1.04	O	−1.05
33	Ti	1.80	Ti	1.71
34	Ti	1.80	Ti	1.58
35	Ti	1.80	Ti	1.73
36	Ti	1.80	Ti	1.90
37	Ti	1.80	Ti	1.77
38	Ti	1.80	Ti	1.65
39	Ti	1.80	Ti	1.47
40	Ti	1.80	Ti	1.82

**Table 5 molecules-28-07134-t005:** The calculated effective mass for electrons and holes at the band edges.

Model	Electrons in the CB		Holes in the VB		IB	
	$m_{e}^{*}$	Direction	$m_{h}^{*}$	Direction	$m_{e}^{*}/m_{h}^{*}$	Direction
CTO	0.069	M-G	0.084	M-G	-	-
LCTO-0.25	0.063	M-G	0.086	M-G	0.070	M-G
CCTO-0.25	0.065	M-G	0.116	M-G	0.073	M-G
LCTO-0.375	0.079	G-R	0.089	G-R	0.141	G-R
CCTO-0.375	0.130	M-G	0.161	M-G	0.257	M-G
LCON2	0.074	M-G	0.149	M-G	-	-
CCON2	0.081	M-G	0.158	M-G	-	-
LCOS1	0.145	M-G	0.198	M-G	0.114	M-G
CCOS1	0.137	M-G	0.196	M-G	0.108	M-G
LCOS1-0.25	0.050	G-R	0.119	G-R	0.087	G-R
CCOS1-0.25	0.095	M-G	0.137	M-G	0.115	M-G
LCOS1-0.375	0.137	M-G	0.146	M-G	0.164	G-X

**Table 6 molecules-28-07134-t006:** Summary of calculated photocatalytic properties of different perovskite-based photocatalysts with different doping strategies.

Doping Type	Functional	Lattice Parameters (nm)	$E_{g}$ (eV)	Maximum Absorption Wavelength (nm)	Refs.
Pd-doped	LDA	a = 7.75, b = 7.75, c = 7.75	0.84	425	[79]
N-doped	GGA-PBE	a = 7.79, b = 7.79, c = 8.02	2.92	-	[60]
S-doped	GGA-PBE	a = 8.37, b = 8.37, c = 8.01	2.55	-	[60]
(La-S)-codoped	GGA-PBE	a = 8.21, b = 7.79, c = 7.83	0.96	412	This work
(Mo-P)-codoped	GGA-PBE	-	1.49	425	[8]
(P-V)-codoped	GGA-PBE	a = 7.82, b = 7.82, c = 7.80	0.95	445	[47]

## Data Availability

All the data are available within the manuscript. Additional data will be provided upon request from the corresponding authors.

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
