# Peer review of "Engineering the Electronic Structure towards Visible Lights Photocatalysis of CaTiO3 Perovskites by Cation (La/Ce)-Anion (N/S) Co-Doping: A First-Principles Study"

_molecules, 2023, doi:10.3390/molecules28207134_

Round 1
Reviewer 1 Report
JOURNAL: Molecules
TITLE: Engineering the Electronic Structure Towards Visible Lights Photocatalysis of CaTiO3 Perovskites by Cation (La/Ce)-Anion (N/S) Co-doping: A First-Principles Study
The authors of the presented theoretical research on the Electronic Structure Towards Visible light photocatalysis of CaTiO3 Perovskites by Cation (La/Ce)-Anion (N/S) Co-doping by using some theoretical approaches. Their calculations are okay, and the outcomes are commended by the authors systematically. It is expected that it will be a practical roadmap to improve new research and developments about the applications. However, some points as stated below require more details. It is suggested to express the originality and differences compared to the with open literacy. Further, it is suggested to scrutinize the effect of the preparation techniques and calculations on the Electronic Structure Towards Visible light photocatalysis of CaTiO3 Perovskites by Cation (La/Ce)-Anion (N/S) Co-doping, etc, Furthermore, it is suggested to extend the introduction in terms of the technological importance and physical/chemical characteristics of the above-mentioned layers. Moreover, some sections need to be improved according to the given literacy and ref. As given below. So, after fulfilling the following issues sufficiently, I recommended that this study be considered for possible publication in molecules.
1-It will be a good advantage to add the graphical abstract and research highlights.
2-Compare the used theoretical models with other possible models related to the presented study. And explain why the present models have been used.
3-The introduction needs to be enlarged according to the advantage of the used CaTiO3 Perovskites preparation techniques over the other conventional techniques as well as the physical properties and advantage of the La or Nd-based perovskite including their technological importance over the other similar systems. Additionally, the impact of the grain size, strain/stress, and boundaries should be considered in more detail. Moreover, explain why CaTiO3 has been chosen among all perovskites and especially La/Ce and N/S have been selected. For that, the following have been recommended for mentioning about them within the text: (i) Chem. Phys. Lett. 722, May 2019, Pages 44-49;(ii) Journal of Alloys and Compounds, 158734, 2021;(iii) Powder Technology, 388, 2021, 274-304,; (v) Optical Materials 133, 112984, 2022; (vi) Beilstein J. Nanotechnol. 2018, 9, 671–685;(vii) J Mater Sci: Mater Electron (2023) 34:873;
4-The obtained results should be commended in more detail with many supportive related refs.
5-The variations in optical absorbance in low or high wavelengths need to be addressed in more detail. also according to it, there are two absorbance edges representing the possibility of secondary phases which should be denoted accordingly.
6-If possible, make a relationship among the obtained results with reasonable comments and facts accordingly.
7-Tabulate all obtained results and compare them with related theoretical calculations or results on the perovskites. And discuss the advantages and disadvantages of the presented models over the other conventional models.
8-Which dopants are best according to your calculations? And for the considered photocatalysts, which type of light is convenient for them and why explain accordingly. This will be very useful for the next researchers.
Reviewer 2 Report
The manuscript is well written and the work is sound and detailed.
The authors need to explain the use of CASTEP as a choice for DFT calculations. The authors also need to explain the choice of 1.3eV as the scissor operator. Were other values tried? Why or why not? The comparison between this and the experimental work the authors mention should be clearly pointed out with the choice of the 1.3 eV value.
Lastly, the introduction, while outstanding, is a little long. The first few sentences in the introduction can easily be removed or significantly reduced to increase the quality of the manuscript readability.
Reviewer 3 Report
Authors investigated the effect of cation-anion doping in CaTiO3 perovskite using a computational approach. The work turned out to be excellent, and very effective in recognising the effects of the substitutions here presented.
To improve even more this paper, I suggest to the authors to combine the electron density analysis with a short Bader's topological analysis of the electron density. In conclusion, I suggest the publishing of the paper in its present form.
Round 2
Reviewer 1 Report
The authors of the revised text have done very well revisions with acceptable, highly enough, and descriptive explanations. Therefore, I recommended the acceptance of the revised text within the journal.